# Surveying the Oral Drug Delivery Avenues of Novel Chitosan Derivatives

**DOI:** 10.3390/polym14112131

**Published:** 2022-05-24

**Authors:** Iyyakkannu Sivanesan, Shadma Tasneem, Nazim Hasan, Juhyun Shin, Manikandan Muthu, Judy Gopal, Jae-Wook Oh

**Affiliations:** 1Department of Bioresources and Food Science, Konkuk University, Seoul 143-701, Korea; isivanesan@gmail.com; 2Department of Chemistry, Faculty of Science, Jazan University, Jazan 45142, Saudi Arabia; sthaque@jazanu.edu.sa (S.T.); hhasan@jazanu.edu.sa (N.H.); 3Department of Stem Cell and Regenerative Biotechnology, Konkuk University, Seoul 143-701, Korea; junejhs@konkuk.ac.kr; 4Department of Research and Innovation, Department of Biotechnology, Saveetha School of Engineering, Saveetha Institute of Medical and Technical Sciences (SIMATS), Thandalam, Chennai 602105, Tamil Nadu, India; bhagatmani@gmail.com (M.M.); jejudy777@gmail.com (J.G.)

**Keywords:** chitosan, chitosan derivative, drug delivery, oral delivery, biomedical applications

## Abstract

Chitosan has come a long way in biomedical applications: drug delivery is one of its core areas of imminent application. Chitosan derivatives are the new generation variants of chitosan. These modified chitosans have overcome limitations and progressed in the area of drug delivery. This review briefly surveys the current chitosan derivatives available for biomedical applications. The biomedical applications of chitosan derivatives are revisited and their key inputs for oral drug delivery have been discussed. The limited use of the vast chitosan resources for oral drug delivery applications, speculated to be probably due to the interdisciplinary nature of this research, is pointed out in the discussion. Chitosan-derivative synthesis and practical implementation for oral drug delivery require distinct expertise from chemists and pharmacists. The lack of enthusiasm could be related to the inadequacy in the smooth transfer of the synthesized derivatives to the actual implementers. With thiolated chitosan derivatives predominating the oral delivery of drugs, the need for representation from the vast array of ready-to-use chitosan derivatives is emphasized. There is plenty to explore in this direction.

## 1. Introduction

Chitosan, made up of β-(1,4)-*N*-acetyl-glucosamine [1,2,3], is obtained following the deacetylation of chitin. Chitin is found extensively in the exoskeletons of crustaceans and insects and in the cell walls of bacteria and fungi [4]. The quality of chitosan is influenced by the source of chitin, separation method and the degree of deacetylation [5]. The major advantages of chitosan are that it is nontoxic, mucoadhesive, hemocompatible, biodegradable and able to exhibit antioxidant, antitumor, antimicrobial properties. These properties render chitosan a highly attractive biomaterial option. The iconic characteristic of chitosan is that it does not provoke intense inflammation nor induce the body’s immunity. Researchers have confirmed that chitosan with different molecular weights and degrees of deacetylation exhibit low toxicity [6,7,8,9]. The catatonic nature of chitosan gives it its bactericidal and bacteriological properties [10,11]. However, chitosan is not soluble in aqueous solutions, a major disadvantage that limits its widespread application in living systems [12].

Chitosan’s surface adherence comes in handy when delivering useful molecules across mucosal pathways and adsorbs molecules that do not have any affinity for mucus [13]. Chitosan, through its permeation-related attributes, is able to open the tight epithelial junctions [14]. Chitosan also plays a role in coagulation. It accelerates the rate of wound healing by enabling interactions between amino and platelet groups [15]. These hemostatic properties are used with respect to wound healing applications. As a material for wound dressing, chitosan possesses chemoattraction, macrophage and neutrophil activation, analgesic properties, acceleration of granulation tissue/re-epithelization, limited scar formation and contraction, hemostasis and antibacterial properties [16]. The antitumor properties of chitosan and its derivatives have been well demonstrated in both in vitro and in vivo models [17]. The beneficial effects of antioxidants are well known [18], chitosan and its derivatives are able to scavenge free radicals in vitro [19,20]. The biodegradability of chitosan is yet another unique feature in biological organisms. Within the system, chitosan interacts with bioenzymes to depolymerize. The degradation breakdown products, *N*-acetyl glucose and glucosamine, are nontoxic to the human body. These degraded intermediates do not stay in the body and have no immunogenicity.

This review focusses on surveying the various novel chitosan derivatives that are available for use as drug delivery options. The milestones achieved based on the use of chitosan derivatives in the area of oral drug delivery has been comprehensively reviewed. The lack of implementing the various chitosan derivatives for oral drug delivery has been highlighted. The plausible reasons for this gap in the application of the various chitosan derivatives for oral drug delivery has been discussed. The possible accomplishments that could be achieved through utilization of the available resources has been addressed under future perspectives.

## 2. Comprehensive List of Novel Chitosan Derivatives

This section deals with a brief overview of the various chitosan derivatives that have been synthesized and are available for biomedical applications. The synthesis and their characterization and their applications have been elaborately reviewed by various authors [21,22,23]; here, we are restricted to a snapshot of their names. Figure 1 gives an overview of the various modification processes involved in the making of various chitosan derivatives.

*N*-(Aminoalkyl) Chitosan is a broad category of chitosan derivatives, which house many other forms. The encapsulation of calcium alginate beads with poly(L-lysine) (PLL), is the most accomplished encapsulation system for sustained delivery of bioactive agents. However, due to its high cost, large scale usage of this system for oral vaccination of animals is not possible. This is why a more economic and reliable microencapsulation chitosan and alginate system was sought after. Succinyl, Quateraminated, and Octanoyl Chitosan Porous chitosan microspheres for the delivery of antigens have been reported by Mi et al. [24]. The porous chitosan microspheres were chemically modified incorporating carboxyl, hydrophobic acyl, and quaternary ammonium groups.

Mitomycin C Conjugated *N*-succinyl Chitosan is the other class of chitosan derivatives. *N*-succinyl-chitosan, due to its carboxyl groups, has low toxicity, excellent biocompatibility and is retained in the body as a drug carrier for prolonged periods. This the reason why highly succinylated succinyl-chitosan (degree of succinylation: [25,26] can be dissolved in alkaline aqueous media, whereas chitosan cannot [27]. Succinyl-chitosan can react easily owing to the –NH_2_ and –COOH groups.

The *N*-Alkyl and Acylated Chitosan derivatives, which greatly benefit from the introduction of an alkyl or acyl chain, contribute greatly to chitosan’s molecular design. This modification of chitosan with hydrophobic branches, improved its solubility properties [28,29]. The introduction of an alkyl chain to water soluble modified chitosan (*N*-methylene phosphonic chitosan) enabled the co-existence of hydrophobic and hydrophilic branches [30]. The alkyl groups in *N*-lauryl-*N*-methylene phosphonic chitosan weaken its hydrogen bonds and provide good solubility in solvents. Holding amphiphilic properties, which are typical for surfactants, this derivative has prospective demands in pharmaceutical and cosmetic fields.

Chitosan hydrochloride derivatives have been demonstrated for their effective in vitro release of ofloxacin from mucoadhesive erodible ocular inserts and ocular pharmacokinetics [31].Thiolated chitosans are obtained by the modification of chitosan with 2-iminothiolane [32], in order to improve the properties of chitosan as excipients in drug delivery systems. Chitosan-2-iminothiolane was obtained by grafting 2-iminothiolane onto the chitosan backbone. This exhibits excellent in situ gelling properties and improved mucoadhesive and drug releasing properties due to the thiol groups on chitosan. Phosphorylated chitosan, which is prepared by reacting chitosan with orthophosphoric acid and urea in DMF [33] or phosphorous pentoxide in methanesulphonic acid, is a water-soluble derivative of chitosan with huge potential for drug delivery.

MCC and SNOCC chitosan derivatives are a biomedically significant class. Mono-N-Carboxymethyl Chitosan (MCC), is a polyampholytic chitosan derivative, soluble at both neutral and alkaline pH [34], synthesized using glyoxolic acid in chitosan [34]. These derivatives are highly soluble and applicable for the administration of polyanionic drugs. It has also been demonstrated by the same group that MCC can improve low molecular weight heparin (LMWH) transport through Caco-2 cells.

Anionic chitosan derivatives were also attempted. N-sulfonato-N,O-carboxymethylchitosan (SNOCC) was produced [35], which retains around 50% of its nitrogen centers on the glucose subunits as free amino groups [36], which contribute to its unique biomedical characteristics.

PEGylated Chitosans are a prominent group of derivatives. Chitosan-PEG for oral peptide delivery was attempted by Prego et al. [37]. PEGylation of chitosan is apt for oral peptide/protein delivery, because generally PEGylation improves biocompatibility [38] and improves stability in GI fluid [37]. PEGylated chitosan showed enhanced solubility of hydrophobics.

## 3. Oral Drug Delivery by Chitosan Derivatives

Although drug delivery is a broad terminology, which is backed up by enumerable reviews when it comes to chitosan and drug delivery and good number of reviews when it comes to chitosan derivatives, this review chooses to specifically delve into oral drug delivery applications. The sections below consolidate what has been achieved in the area of oral drug delivery based on chitosan derivatives and micro/nano particulate chitosan.

### 3.1. Chitosan/Chitosan Derivatives

When drugs are administered orally, they must be able to survive various ranges of pH and gastrointestinal tract (GIT) secretions. The very process of oral drug absorption rests on transport (via passive diffusion, carrier-mediated transport, or pinocytosis) across the GIT membrane. This is impacted by various GIT physiological. The oral mucosa has a thin epithelium and rich vascularity, which is makes it ideally fit for buccal and sublingual administration [39]. The release of drugs from chitosan and its derivatives follows the conventional protocol that holds good for chitosan. Drug release is influenced by the hydrophilicity of chitosan and pH of the swelling solution. The chitosan-drug release mechanism involves swelling, diffusion of drugs through the polymeric matrix and polymer erosion [40] (Figure 2).

Figure 3 lists the limitations that chitosan and its derivatives have broken, when it can to oral drug delivery. Drug delivery via the oral route is the easiest and e most convenient for patients. Chitosan because of its mucoadhesive nature, is able to protect labile drugs from GIT enzymatic degradation. Additionally, it is able to enhance absorption of administered therapeutic agent without affecting the biological system. This makes chitosan a valuable candidate as an oral delivery agent. Not only chitosan, but also chitosan micro-/nanoparticles have been demonstrated for oral drug delivery. Intestinal disinfection, suppression of *Helicobacter pylori* and dealing with ulcerative colitis, have been accomplished following treatment with antibiotic loaded chitosan particles. Amoxicillin and clarithromycin loaded into chitosan particles inhibited *H. pylori* [41,42]. The mucoadhesive properties of chitosan enabled prolonged delivery and oral bioavailability of acyclovir, an antiviral agent. This was because acyclovir chitosan microspheres could enhance drug retention in the upper GIT [43]. Protection against GIT degradation, improvement of oral bioavailability of insulin and enhancement of bioadhesion, have been reported as a result of its encapsulation into chitosan microspheres [44].

Chitosan-based delivery systems have been applied for the protection of insulin from degradation in the upper GIT. Furthermore, it has been used to carry out the release of insulin at the colon (through degradation of the chitosan glycosidic linkage by colon microflora) [45]. Chitosan microspheres coated with cellulose acetate butyrate, loaded with 5-aminosalicylic acid (5-ASA) to treat ulcerative colitis is reported. Here, the bioadhesive nature of chitosan microspheres comes handy [46]. Another study reported localization of 5-ASA in the colon and low drug systemic bioavailability following oral administration of 5-ASA-loaded chitosan-Ca-alginate microparticles to Wistar male rats [47]. The fact that chitosan is highly soluble in the acidic medium, leading to drug burst in the stomach, has been mitigated using pH-sensitive polymer coatings [48,49,50].

Chitosan derivatives have also been reported for oral delivery of therapeutic peptides and proteins. Unmodified native chitosan itself has been proven for its oral peptide and protein delivery (e.g., capability to open tight junctions, mucoadhesive properties), with this being the case, how much more so with the use of chitosan derivatives. Recently, the potential of certain modified chitosans including TMC [51], thiolated chitosan [52,53] and chitosan-enzyme inhibitor conjugates [54,55,56] for noninvasive gene delivery has been widely reported. In addition, thiolated chitosan is able to inhibit efflux pumps, in particular P-glycoprotein (P-gp). In this way, thiolated chitosan comes handy when it comes to oral delivery of P-gp substrates [57,58,59]. The potential of chitosan, TMC and MCC for oral delivery of vaccine have been previously reviewed [60]. We touch on the highlights of these [61,62] reviews here.

The effect of two different trimethyl chitosans (TMC) on the oral absorption of buserelin, a peptide drug, after intraduodenal administration in rats is reported [63] Both formulations significantly enhanced buserelin plasma levels. Enhanced absorption in the presence of TMC60 (60% trimethylation) is because of the inherent ability of TMC60 to open tight junctions. The impact of TMC solutions on octreotide in vitro permeation and in vivo absorption in rats was also investigated [63]. The intrajejunally administered TMC solution led to a fivefold increase in the absorption of octreotide compared to octreotide standalone. The effect of various liquid formulations on the oral bioavailability of octreotide was studied in pigs [64]. Studies with MCC and SNOCC towards oral delivery of LMWH [34,35], confirmed that chitosan derivatives in a concentration of 3% improved the oral bioavailability of LMWH.

In vivo studies using thiolated chitosan tablets were applied using peptide drugs as well as efflux pump substrates. Enteric coated chitosan–TBA conjugated with salmon calcitonin for the oral administration to rats were tested. Besides chitosan–TBA, the tablets contained two different chitosan–enzyme inhibitor conjugates, (chitosan–BBI conjugate and chitosan–elastatinal) [65]. Oral administration of this chitosan conjugate showed decreased plasma calcium levels for several hours [66]. Another study, where stomach targeted delivery system for salmon calcitonin was investigated using tablets containing chitosan–TBA as well as chitosan–pepstatin [67]. The efficacy of chitosan–TBA/GSH for oral peptide delivery was studied using the peptide drug antide. Antide was not absorbed after oral administration; however, absorption of the drug was reported following oral administration of chitosan TBA/GSH tablets [26]. Besides peptides and proteins, oral bioavailability of efflux pump substrates was improved using thiolated chitosan tablets were used. Oral bioavailability of the P-gp substrate Rhodamine 123 (Rho-123) was reported [59]. Guggi et al., used optimized tablets comprising of chitosan-TBA with lower molecular mass (75–150 kDa instead of 400 kDa) and demonstrated a 5.5-fold increase in Rho-123 AUC in comparison to the Rho-123 buffer solution. Guggi et al. investigated the effect of various calcitonin containing tablets on the blood calcium level of rats after oral administration. Compared to tablets containing calcitonin and chitosan only, marginal reduction of the calcium level was observed after administration of chitosan–pepstatin conjugate tablets [67]. Oral insulin delivery using insulin and chitosan–aprotinin conjugate, showed reduced blood glucose level, 8 h after oral administration [68].

### 3.2. Micro- and Nanoparticulate Oral Drug Delivery Systems Based on Chitosan Derivatives

#### 3.2.1. Microparticulate Chitosan Derivatives Oral Drug Delivery Systems

Authors reported the preparation of liposome microspheres were coated with TMC and chitosan–EDTA. In vivo studies on oral absorption of insulin, confirmed that chitosan EDTA coated liposomes decreased blood glucose [69]. Microspheres based on chitosan–succinate proved their potential for oral delivery of insulin [25]. The delivery system was tested in vivo in diabetic rats, with chitosan–succinate microspheres, the relative pharmacological efficacy showed fourfold improvement [25]. Intragastric administration of calcitonin containing liposomes coated with dodecylated chitosan was confirmed in rats. Similar results were obtained in case of chitosan–phthalate microspheres too. PEGylated chitosan was tested for oral delivery of salmon calcitonin. Alginate–chitosan microspheres with narrow size distribution were prepared by membrane emulsification technique in combination with ion (Ca^2+^) and polymer (chitosan) solidification. The blood glucose level of diabetic rats was effectively reduced. It was made available for as long as 60 h after oral administration of the insulin-loaded alginate–chitosan microspheres. Therefore, the alginate–chitosan microspheres were found to be promising vectors showing a good efficiency in oral administration of protein or peptide drugs [70]. Chitosan microparticles prepared using the precipitation/coacervation method to obtain biodegradable chitosan microparticles. The entrapped ovalbumin was released after intracellular digestion into the Peyer’s patches. The proved that the labeled chitosan microparticles could be taken up by the epithelium of the murine Peyer’s patches. Since uptake by Peyer’s patches is an essential step in oral vaccination, these results confirmed that the chitosan microparticles are useful when it comes to vaccine delivery system [71]. Chitosan and chondroitin sulphate microspheres were prepared and reported for controlled release of metoclopramide hydrochloride in oral administration [72]. Microparticles prepared by ionic crosslinking between tripolyphosphate (TPP) and chitosan (Cs) were applied to enable the oral bioavailability of curcumin. The developed microparticles are reported to successfully enhance the dissolution of the poorly water-soluble drug Cur, and eventually, improve its oral bioavailability effectively [73].

#### 3.2.2. Nanoparticulate Chitosan Derivatives Oral Drug Delivery Systems

TMC-based insulin-loaded nanoparticles were investigated, it was reported that insulin-TMC polyelectrolyte complexes exhibited higher colloidal stability in simulated intestinal fluid and protected insulin from trypsinic degradation [74]. TMC nanoparticles has also been demonstrated for its oral vaccine delivery. Intragastrical (IG) administration of TMC-nanoparticles containing the model vaccine urease could result in higher IgG and IgA levels [75]. Another study reported the efficiency of TMC as vector for in vitro and in vivo gene delivery [76]. Three different TMC-based nanoparticles encapsulated pDNA encoding green fluorescent protein (GFP) were demonstrated for their successful delivery attributes. Nanoparticles based on chitosan–TGA and pDNA for oral delivery are also reported [53]. Acrylic nanoparticles with chitosan–TBA are also reported. In vivo studies with thiolated chitosan nanoparticles for oral delivery are still lacking, however, oral insulin delivery using thiolatedpoly(acrylic acid) nanoparticles [77] and intranasal gene delivery using chitosan-TGA nanoparticles have been demonstrated [52]. Fucoidan (FD) has hypoglycemic effects, TMC and FD were loaded with insulin. TMC/FD NPs are pH sensitive and defend insulin from degradation in the GIT. Moreover, they enhance the cellular transport of insulin across the intestinal barrier [78]. The delivery of insulin via glycerol monocaprylate-modified chitosan nanoparticles has also been demonstrated using TMC/FD NPs [79]. A nanoemulsion was coated with two different PEGylated chitosans. In vivo studies in rats showed, that the oral uptake of salmon calcitonin when administered in carriers coated with PEGylated chitosan was higher than the nanoemulsion alone [37]. Table 1 gives the consolidated list of chitosan derivatives that have been employed for oral drug delivery applications.

## 4. Future Endeavors

This review briefly surveyed the current scenario of oral drug delivery using chitosan derivatives. Drug delivery is a very appropriate subject area, which chitosan have enormously impacted. We ran a pubmed search, using keywords, chitosan and drug delivery, chitosan derivatives and oral drug delivery, chitosan derivatives and drug delivery. Backed up by a total of 10,000 odd publications as per our pubmed search from 1981–2022, chitosan has indeed generously contributed to drug delivery. Novel chitosan derivatives, which are the second-generation innovations emerging from chitosan, have a 2635 publication record when it comes to drug delivery applications.

Chitosan derivatives are well reported for their use in delivery of poorly soluble drugs, for colon-targeted drug delivery, for mucosal drug delivery, ocular drug delivery and topical delivery [81,82,83,84].

Chopra et al. [85] have extensively reviewed the advances and potential applications of chitosan derivatives as mucoadhesive biomaterials in modern drug delivery. When it comes to drug delivery, the drawbacks of chitosan have been overcome through derivatives such as carboxylated, various conjugates, thiolated, and acylated chitosan and Tan et al. have reviewed the applications of quaternized chitosan as antimicrobial agents, including their antimicrobial activity, mechanism of action and biomedical applications in orthopedics [86]. These have become an appropriate platform for sustained release at a controlled rate, prolonged residence time, improved patient compliance through reduced dosing frequency, enhanced bioavailability leading to significant improvement in therapeutic efficacy.

Currently, chitosan derivative nanoparticles are mainly used for sustained release, preparation of targeted drugs and as vectors for gene therapy. As delivery carriers, chitosan and its derivatives are usually available as microspheres, nanoparticles, micelles, and gels in delivery carriers [87,88]. Besides these options, chitosan derivative nanoparticles are also used for the delivery of polypeptides. Chitosan derivative nanoparticles interact with peptides through strong hydrogen bonds and static electricity, obtaining peptide-loaded nanoparticles. Fatty-acid-modified quaternary ammonium chitosan nanoparticles loaded with insulin have been shown to be beneficial [89]. Chitosan derivative nanoparticles have also been applied for gene delivery. Gene therapy is a promising strategy for challenging diseases. A key step in gene therapy is the successful delivery of genes [90,91]. Chitosan derivative nanoparticles, as non-viral vectors, have excellent solubility, biodegradability, biocompatibility, non-toxicity and a higher transfection rate than chitosan nanoparticles [92]. Methoxy polyethylene glycol-modified trimethyl chitosan (mPEG-TMC) has been covalently linked to doxorubicin (DOX) and cis-itaconic anhydride (CA), for better anti-tumor activities [93,94]. O-carboxymethyl chitosan inhibited tumor cell migration in vitro [95]. The poly-β-amino ester nanoparticle loading gene, after the addition of thiolated O-carboxymethyl chitosan, showed a higher cell transfection rate [96]. These are a notable few brief mention of the drug delivery potentials of chitosan derivatives, which have been dealt in detail by earlier reviews.

Chitosan derivatives upgraded to break many of the limitations that chitosan was facing, and with that reputation, it was believed that higher research curiosity and much more research interest would be evident. This expectation is well below the actual trend. As for oral drug delivery, chitosan derivatives are within the 500-article mark, which is one fifth lesser than the interest on chitosan derivatives and drug delivery. Figure 4 summarizes this trend. However, as this review points out, there is definitely a high potential contribution from chitosan derivatives in biomedical applications and drug delivery, which we stress has not been fully tapped into in terms of oral drug delivery applications. This review hopes to provoke some though and awareness towards this area of research.

Non-invasive oral drug delivery is the crown of drug delivery approaches, chitosan derivatives are the latest generation upgrades, a fusion of both these should break numerous boundaries and limitations. The fact that this is truly an interdisciplinary area, where synthetic chemists and pharmacologist need to collaborate to access the full potential of either expertise, may be the retardant. The reason for the low enthusiasm could be the interdisciplinary nature of this area of research. There is no dearth for chitosan derivatives, as pointed out by the review, diverse chitosan derivatives are in the market. Yet, as pointed out in this review, only thiolated chitosans have been predominantly applied, and few other scattered versions too. There are a whole lot of options to consider and avenues that they would open up which are yet to be looked into. This review hopes to enthuse the researchers in this direction.

Combining nanoaspects of chitosan with synthesis of chitosan derivatives is definitive progress in this area. Nanoforms have always pushed limitations of various applications, and there is surely a lot more to derive from nanostucturization of the chitosan derivatives. Oral drug delivery has benefitted greatly from the use of nanochitosan forms; combining chitosan derivatives with nano aspects could prove highly beneficial.

## 5. Conclusions

The objective of this review was to showcase the wealth of available chitosan derivatives and to evaluate their achievements in the area of oral drug delivery. Numerous reviews exist in the area of chitosan and drug delivery, chitosan derivatives and drug delivery applications are also well reported. We reviewed the comparatively less-reported chitosan derivative application into oral drug delivery. During the review process, it became clear that there is no doubt as to the advantages of employing the use of chitosan derivatives for oral drug delivery purposes. However, as pointed out in the review, there is a huge gap between the available knowledge and the synthesized chitosan derivations and their oral drug delivery applications. There are so many derivatives synthesized, yet only few have been used for oral drug delivery applications. The reasons for this gap and the various reasons that could have led to this have been speculated. The need to bridge these ends have been emphasized. There is definitely much to harness and more to achieve, through proper inclusion of chitosan derivatives that have so far not been attempted for oral drug delivery applications.

## Figures and Tables

**Figure 1 polymers-14-02131-f001:**
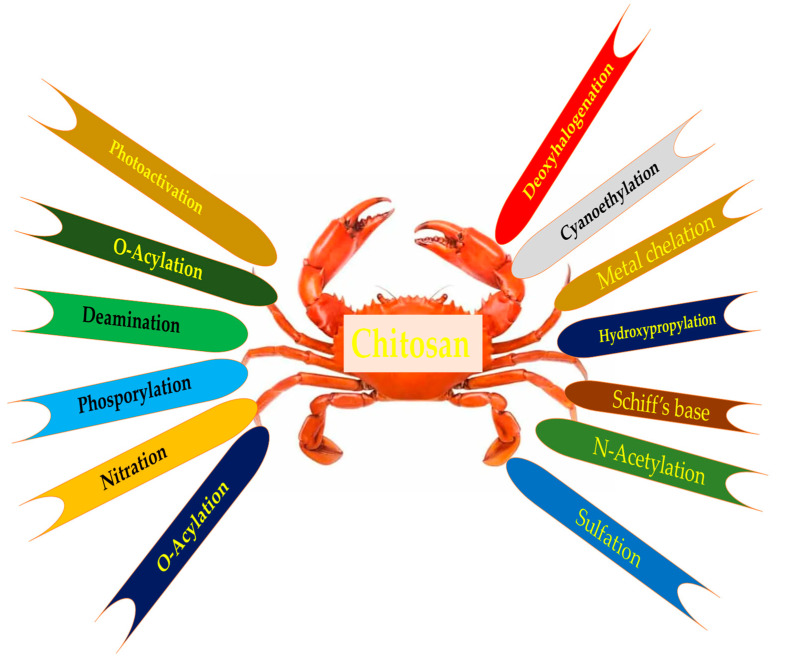
Overview of various chemical modification processes that go into the making of chitosan derivatives.

**Figure 2 polymers-14-02131-f002:**
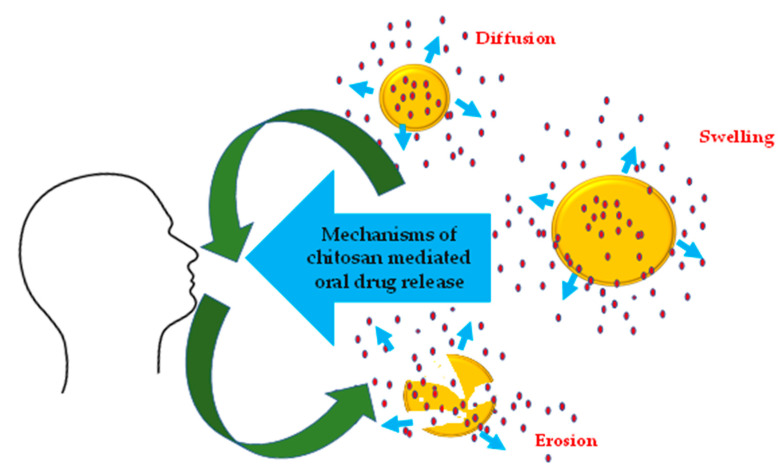
Mechanisms of drug release in chitosan and its derivatives.

**Figure 3 polymers-14-02131-f003:**
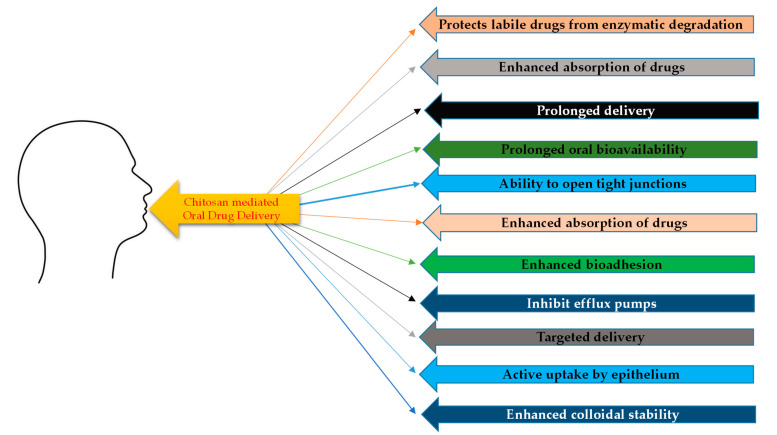
Listings of the limitations in oral drug delivery that chitosan and its derivatives have helped overcome.

**Figure 4 polymers-14-02131-f004:**
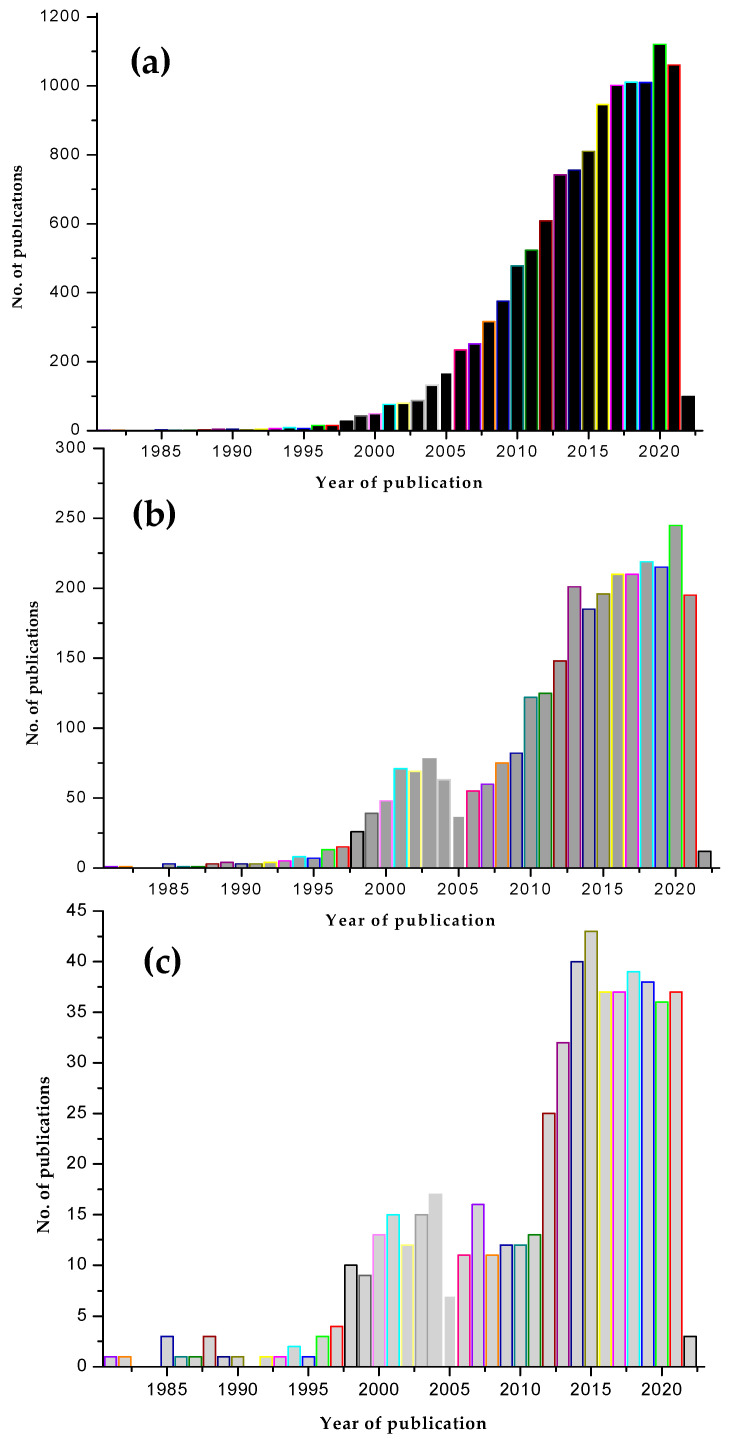
Comparative bar diagrams of research articles published in the area of (**a**) chitosans and drug delivery (**b**) chitosan derivatives and drug delivery (**c**) chitosan derivatives for oral drug delivery, based on our pubmed search.

**Table 1 polymers-14-02131-t001:** Chitosan derivatives that have been used for oral drug delivery applications.

Chitosan Derivative	Oral Drug Delivery Application	References
TMC, thiolated chitosan	noninvasive gene delivery	[52,53]
Thiolated chitosan	oral delivery of P-glycoprotein (P-gp) substrates	[57,58,59]
TMC, MCC	oral vaccine delivery	[60]
Trimethyl chitosans (TMC)	oral absorption of the peptide drug buserelin after intraduodenal administration in rats	[63]
TMC	octreotide in vitro permeation and in vivo absorption in rats	[80]
TMC	oral bioavailability of octreotide in pigs	[64]
MCC, SNOCC	oral delivery of LMWH	[34,35]
Chitosan–TBA	oral administration of drug salmon calcitonin to rats	[65]
Chitosan–TBA/chitosan–enzyme inhibitor conjugate	Delivery of drug salmon calcitonin	[66]
Chitosan–TBA, chitosan–pepstatin	Stomach targeted delivery of salmon calcitonin	[67]
Chitosan–TBA/GSH	oral peptide delivery of peptide drug antide	[26]
Chitosan–TBA	Oral bioavailability of the P-gp substrate Rhodamine 123	[59]
Chitosan–TBA	5.5-fold increase in Rho-123 AUC	[59]
Chitosan–pepstatin conjugate tablets	Reduction of blood calcium level of rats after oral administration	[67]
Chitosan–aprotinin conjugate	Oral insulin delivery	[68]
TMC–nanoparticles–Vaccine urease	Oral vaccine delivery–higher IgG and IgA levels	[75]
TMC nanoparticles	gene delivery	[76]
TMC–based nanoparticles encapsulatde pDNA encoding green fluorescentprotein (GFP)	Oral delivery	[53]
chitosan-TGA and pDNA	Oral delivery	[53]
thiolatedpoly(acrylic acid) nanoparticles	Oral insulin delivery	[77]
Chitosan–TGA nanoparticles	intranasal gene delivery	[52]
liposomes coated with dodecylated chitosan	Intragastric administration of calcitonin	[37]
TMC and chitosan–EDTA	Oral absorption of insulin	[78]
chitosan-succinate	oral delivery of insulin chitosan-succinate microspheres	[25]
Chitosan–succinate microspheres	in vivo insulin delivery in diabetic rats	[25]
TMC/FD NPs	defend insulin from degradation in the GIT, enhance transport	[79,80]

## Data Availability

Not applicable.

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
