# Peer review of "Surveying the Oral Drug Delivery Avenues of Novel Chitosan Derivatives"

_polymers, 2022, doi:10.3390/polym14112131_

Round 1

Reviewer 1 Report

The paper eentitled "Surveying the oral drug delivery avenues of novel chitosan derivatives" by Iyyakkannu Sivanesan and co, illustrates some chitosan derivatives with applications in oral drug delivery. 

The subject is intersting but the paper is not suitable for publications in this form because:

1) The reason of writting this review it is not clear because the latest works in the field are not presented (few papers from the last 3 years are cited) and the purpose of the paper is not clear.

2) The table 1 does not reflect any idea and the bibliographic referecnes are missing. Even the format of the yable is wrong. Why those chitosan derivatives are listed from others, apart they have been used for oral drug delivery applications. 

3) The conclusion is vague, does not reflect anything new or original- as "The objective of this review was to showcase the wealth of available chitosan derivatives and to evaluate their achievements in the area of oral drug delivery. During the review process, it became clear that there is no doubt on the advantages of employing the use of chitosan derivatives for oral drug delivery purposes"

Author Response

The paper eentitled "Surveying the oral drug delivery avenues of novel chitosan derivatives" by Iyyakkannu Sivanesan and co, illustrates some chitosan derivatives with applications in oral drug delivery. 

The subject is intersting but the paper is not suitable for publications in this form because:

1) The reason of writting this review it is not clear because the latest works in the field are not presented (few papers from the last 3 years are cited) and the purpose of the paper is not clear.

Ans. We would like to thank the reviewer for the revision opportunity. We have now revised the paper based on your comments. We have added the necessary citations. Thank you. 

2) The table 1 does not reflect any idea and the bibliographic referecnes are missing. Even the format of the table is wrong. Why those chitosan derivatives are listed from others, apart they have been used for oral drug delivery applications. 

Ans. We are extremely sorry, the table had gone out of margin thats why you could not view the references. We have now reformated the table. thank you.  

3) The conclusion is vague, does not reflect anything new or original- as "The objective of this review was to showcase the wealth of available chitosan derivatives and to evaluate their achievements in the area of oral drug delivery. During the review process, it became clear that there is no doubt on the advantages of employing the use of chitosan derivatives for oral drug delivery purposes"

Ans. We have now rewritten the conclusions to be more specific. Sorry about that. Thank you for your kind understanding. 

Reviewer 2 Report

The author mentioned about Chitosan derivative synthesis and practical implementation for oral drug delivery. I feel that in present form the manuscript is suitable to publish in polymers as author write very simple facts. Therefore, I would like to suggest about the revision of the whole manuscript with considering the following points;

In introduction some advantages/disadvantages, background of the topic must be incorporated

Some critical discussion should be there.

Figure related to the mechanism of application in biomedical field must be included.

Avoid to write these very simple sentences like line number 14 , 25 etc.

Author Response

The author mentioned about Chitosan derivative synthesis and practical implementation for oral drug delivery. I feel that in present form the manuscript is suitable to publish in polymers as author write very simple facts. Therefore, I would like to suggest about the revision of the whole manuscript with considering the following points;

In introduction some advantages/disadvantages, background of the topic must be incorporated

Ans. We have added as you have specified. Thank you

Some critical discussion should be there.

Figure related to the mechanism of application in biomedical field must be included.

Ans. We have added a figure. Thank you. 

Avoid to write these very simple sentences like line number 14 , 25 etc.

Ans. Sorry about that, we have now revised the writing style too. Thank you.

Reviewer 3 Report

Jae-Wook Oh and co-workers have reviewed the research work on chitosan derivatives for oral drug delivery applications. They have provided the rationale for the importance and applications of chitosan derivatives for various delivery purposes either orally or with other means. They started their review by providing a brief list of chitosan derivatives. Subsequently, the oral drug delivery by chitosan derivatives is given. In the short review, the particle-based delivery is shown. 

Despite the topic and chitosan-based biomaterials being very promising in biomedical applications, the current review lacks any novelty or detailed report of up-to-date chitosan-based materials for drug delivery applications. 

I found the current version of a Review paper is not suitable for publication in its current form. However, authors may consider comprehensive revision and re-submission to Polymers. 

First of all, the writing is very unorganized. 

There are lots of spelling mistakes such as the third paragraph of section 2 on page 2 (ita written instead of its), (inulin written instead of insulin on the second line from the bottom on page 4), etc.

The future direction is not impressive, it can be improved.

Chitosan is a very promising biomaterial and its future impact in the biomedical field is very promising, which is not reflected in this review article.

The current reference list is good but can be improved in revision.

Authors may consider some direct examples in the figure forms. 

Once again, I am happy to see this topic in Review, but the current version doesn't give any details or novelty. If the authors consider the above suggestions during major revision, then I am optimistic about this paper to be appear in the journal Polymers. 

Author Response

Jae-Wook Oh and co-workers have reviewed the research work on chitosan derivatives for oral drug delivery applications. They have provided the rationale for the importance and applications of chitosan derivatives for various delivery purposes either orally or with other means. They started their review by providing a brief list of chitosan derivatives. Subsequently, the oral drug delivery by chitosan derivatives is given. In the short review, the particle-based delivery is shown. espite the topic and chitosan-based biomaterials being very promising in biomedical applications, the current review lacks any novelty or detailed report of up-to-date chitosan-based materials for drug delivery applications. 

Ans. The novelty of this review is the fact that we have compiled what has been achieved in the area of chitosan derivatives and oral drug delivery. This is a specific targeted review. Generalized reviews on this subject area are abundant, chitosan and drug delivery for one is a very well addressed subject area. The reviewers should note that we have concentrated on chitosan derivatives and not drug delivery but specifically oral drug delivery. There is nothing much published in this direction. Thank you for your kind understanding. 

I found the current version of a Review paper is not suitable for publication in its current form. However, authors may consider comprehensive revision and re-submission to Polymers. 

First of all, the writing is very unorganized. 

Ans. We have now organized the flow. Thank you. 

There are lots of spelling mistakes such as the third paragraph of section 2 on page 2 (ita written instead of its), (inulin written instead of insulin on the second line from the bottom on page 4), etc.

Ans. Very sorry about that, we have now done a thorough check. Thank you. 

The future direction is not impressive, it can be improved.

Ans. Yes we have improved it now. 

Chitosan is a very promising biomaterial and its future impact in the biomedical field is very promising, which is not reflected in this review article.

Ans. We have worked on this part too. 

The current reference list is good but can be improved in revision.

Ans. Improved. Thank you. 

Authors may consider some direct examples in the figure forms. Once again, I am happy to see this topic in Review, but the current version doesn't give any details or novelty. If the authors consider the above suggestions during major revision, then I am optimistic about this paper to be appear in the journal Polymers. 

Ans. Thank you for the optimism, we have worked on the paper and projected the novelty factors of this review. We also have added one figure. Thank you. 

Round 2

Reviewer 1 Report

The paper can be accepted in present form, the errors were corrected

Reviewer 2 Report

Still, there is a need to improve the overall standard of the article in terms of language, typo errors etc.

such as line 97, 228 etc

The formal sentences should standardized accordingly.

Reviewer 3 Report

The paper could be improved even further. But, after the improvement during the Revision, I recommend its acceptance in its present form.